# Growth Temperature Influence on Atomic-Layer-Deposited In_2_O_3_ Thin Films and Their Application in Inorganic Perovskite Solar Cells

**DOI:** 10.3390/nano11082047

**Published:** 2021-08-11

**Authors:** Umme Farva, Hyeong Woo Lee, Ri-Na Kim, Dong-Gun Lee, Dong-Won Kang, Jeha Kim

**Affiliations:** 1Department of Energy Convergence Engineering, Cheongju University, Cheongju 28503, Korea; ummefarva@gmail.com (U.F.); hwkj0224@gmail.com (H.W.L.); kimrina74@gmail.com (R.-N.K.); 2School of Energy Systems Engineering, Chung-Ang University, Seoul 06974, Korea; padawan1215@naver.com

**Keywords:** atomic layer deposition, In_2_O_3_, XPS analysis, electrical properties, electron transport layer, CsPbI_2_Br, perovskite solar cells

## Abstract

Recently, indium oxide (In_2_O_3_) thin films have emerged as a promising electron transport layer (ETL) for perovskite solar cells; however, solution-processed In_2_O_3_ ETL suffered from poor morphology, pinholes, and required annealing at high temperatures. This research aims to carry out and prepare pinhole-free, transparent, and highly conductive In_2_O_3_ thin films via atomic layer deposition (ALD) seizing efficiently as an ETL. In order to explore the growth-temperature-dependent properties of In_2_O_3_ thin film, it was fabricated by ALD using the triethyl indium (Et_3_In) precursor. The detail of the ALD process at 115–250 °C was studied through the film growth rate, crystal structure, morphology, composition, and optical and electrical properties. The film growth rate increased from 0.009 nm/cycle to 0.088 nm/cycle as the growth temperature rose from 115 °C to 250 °C. The film thickness was highly uniform, and the surface roughness was below 1.6 nm. Our results confirmed that film’s structural, optical and electrical properties directly depend on film growth temperature. Film grown at ≥200 °C exhibited a polycrystalline cubic structure with almost negligible carbon impurities. Finally, the device ALD-In_2_O_3_ film deposited at 250 °C exhibited a power conversion efficiency of 10.97% superior to other conditions and general SnO_2_ ETL.

## 1. Introduction

Indium oxide, In_2_O_3_, has been widely considered in numerous applications in the fields of optics, photovoltaic devices, thin-film transistors, sensors, and transparent windows in liquid crystal displays [1,2,3] because of its excellent electrical properties as well as high transparency in the visible spectrum [4,5]. In addition, bulk In_2_O_3_ has high carrier mobility (140–170 cm^2^ V^−1^ s^−1^), low resistivity, and excellent thermal and chemical stability [6]. Perovskite solar cells (PSCs) have gained extensive attention over the past decade because of their recent increased power conversion efficiency (PCE) from 3.8% (2009) to certified 25.2% PCE [7] in a short span. As a result, it has been widely accepted and has emerged as the fastest-growing photovoltaic technology of the current era. In device architecture, wide bandgap, high electron mobility are required in those materials that are appropriate for ETL. This ETL plays a critical role in effectively extracting the electrons, and efficiently transporting photogenerated electrons from perovskite to ETL at the ETL/perovskite interface constrains charge recombination. In planar PSCs, a thin, compact TiO_2_ layer is typically used as inorganic ETL and achieved remarkable PCEs [8]. For that, an excellent TiO_2_ film requires thermal treatment at high temperatures of ≥500 °C, which is incongruous for the fabrication for flexible substrate, including reducing PSCs’ stability due to its photocatalytic effect [9].

Recently, In_2_O_3_ has been put forward as a propitious alternative ETL in PSCs [10,11] because of its wide bandgap energy (3.5–3.75 eV), deep conduction band (CB) with good energy level alignment and high stability. Moreover, its immense electrical properties, comprehensive optical transparency, and antireflective nature make it a prominent ETL material. Unfortunately, there are hardly any reports on In_2_O_3—_based PSCs available. The sol-gel-fabricated In_2_O_3_ ETL shows a power conversion efficiency of 14.83% with hysteresis and a similar method related to upgradation up to 15.3% [10,12]. 

However, spin-coating In_2_O_3_ films exhibited 14.63% PCE while a low-temperature combustion solution process had 18.12% PCE [13,14]. These can be challenging as these methods have some drawbacks, including poor morphology with a high density of pinholes in In_2_O_3_ due to vigorous hydrolysis of In^3+^. Apart from this, the above process requires annealing at high temperatures (>300 °C) to remove the impurities and obtain highly conductive, crystalline In_2_O_3_ film, which is harmful to a flexible substrate and maximizes the chance of recombination in devices. Along with this, so far, inorganic perovskite materials have not to be employed on In_2_O_3_ ETL for PSCs. Consequently, these issues have paid substantial interest in establishing a facile method to grow pinhole-free compact In_2_O_3_ films with controlled optical and electrical properties for organic or inorganic perovskite solar cells. 

Currently, several growth techniques have been conducted to develop the deposition process, including spray pyrolysis [15], sol-gel, dc magnetron sputtering [16], electron beam evaporation [17], pulsed laser deposition [18], and ALD to obtain good quality In_2_O_3_ films. The ALD method has received considerable attention because it can produce high-quality, dense films with atomically precise coatings on various surfaces [19]. A sub-nanometer film thickness and composition transmit light and transport carriers without any optical and electrical loss. Additionally, ALD’s chemical reactions form reliable covalent linkages between the ALD film and the underlying substrate can significantly enhance the stability of deposited film [20].

Notably ALD precursors require some peculiar characteristics such as high reactivity, sufficient vapor pressure, and good thermal stability. So far, indium precursors such as halogenated [21,22], metalorganic [23,24,25], organometallic [26,27,28,29], or indium compounds [30] have been extensively utilized for ALD In_2_O_3_ deposition with restricted attainment. Halogenated precursors required a high growth temperature of 300–500 °C and have low pressure, and ALD reaction byproduct HCl, which is corrosive, can make equipment deteriorate. While metalorganic precursors can be used in low-temperature ALD windows, they are at the same time solid precursors, have some limitations as they have a high probability of particle contamination. Moreover, they are not suitable for industrial application. The organometallic liquid indium precursors are also employed for the ALD process, but the result revealed low reactivity and steric hindrance due to the indium precursor’s bulk size [23]. 

Previously ALD process uses commercially available liquid indium precursor triethylindium (Et_3_In) and O_3,_ which was reported at relatively low deposition temperature (50–250 °C), showed the high reactivity of Et_3_In with O_3_ [31]. However, no further study exists yet with this novel precursor as it exhibited high-density good electrical performance film among other indium precursors studied in the same report. It is evident that the film morphology, crystallinity, chemical composition, and existing impurity lessens the optical and electrical properties of the ensuing In_2_O_3_ film. 

Therefore, in this work, we investigated a device-grade In_2_O_3_ thin film with deposition temperatures from 115 °C to 250 °C with control growth per cycle, performing a detailed study of film growth by terms of morphology, crystallinity, chemical composition, binding states, and enhance the electrical and optical properties to improve the seizing efficiently as an ETL for the planar-type inorganic PSCs. Furthermore, the first-time CsPbI_2_Br-based PSCs device performance using ALD-In_2_O_3_ films as ETLs deposited at 150, 200, and 250 °C were fabricated and investigated. 

## 2. Materials and Methods

### 2.1. Film Growth

A commercial six-inch ALD chamber (iSAC Co. Ltd., iOV m100, Daejeon, Korea) was used for film deposition. The In_2_O_3_ thin films were deposited in the different growth temperatures of 115–250 °C on two different types of substrates, Si (100) wafer (15 × 15 mm^2^), (10 × 10 mm^2^), and soda-lime glass (SLG) (10 × 10 mm^2^). All the chemicals were reagent grade and used as received. The substrates were cleaned ultrasonically with acetone, isopropanol, and deionized water for 30 min and then blown dry with N_2_ gas. The substrates were then loaded into the preheated ALD chamber; after that, the substrate was permitted to outgas by Argon (Ar) purge in the ALD chamber for 20 min to clean and avoid contamination. The Et_3_In (iChems. Co., Hwaseong, Korea) precursor was in a stainless steel canister. The bubbler temperature was maintained at 40 °C for obtaining an appropriate vapor pressure for the ALD deposition process. Ar gas (99.99%) was used as a carrier gas having a fixed flow rate of 50 sccm (standard cubic centimeter per minute), and working pressure was maintained at 300 mTorr. An ozone reactant was generated by an ozone generator using O_2_ gas (99.99%). The ALD deposition cycle consists of sequence Et_3_In pulse (0.3 s)-Ar purge (5 s)-O_3_ (2 s) pulse-Ar purge (10 s). The pulse time unit is given in seconds (s). The heating chamber, the substrate, the bubbler temperatures, and the process lines were controlled using a computer, and the gas flows were controlled by using mass flow controllers (MFCs).

### 2.2. Materials for Device Fabrication 

Cesium iodide (CsI; 99.999%, metals basis) and tin oxide (SnO_2_; 15 wt% in an H_2_O colloidal dispersion) were purchased from Alfa Aesar (Seoul, Korea). Lead bromide (PbBr_2_; 99.999%, trace metals basis) was purchased from Sigma Aldrich (Seoul, Korea). Lead iodide (PbI_2_; 99.99%, trace metals basis) was purchased from TCI (Seoul, Korea). Poly[(9,9-dioctylfluorenyl-2,7-diyl)-co-(4,4′-(N-(4-secbutylphenyl)di-phenyl-amine))] (TFB) was purchased from Sigma Aldrich. Poly (3-hexylthiophene-2,5-diyl) (P3HT) was purchased from Rieke Metals (Lincoln, NE, USA). Dimethyl sulfoxide (DMSO; 99.8%) was purchased from Samchun Chemicals (Seoul, Korea). Chlorobenzene (CBZ; 99.8%) was purchased from Kanto Chemical (Tokyo, Japan). The CsPbI_2_Br perovskite precursor solution was prepared by dissolving an equimolar ratio of CsI (1.8 M), PbI_2_ (1.8 M), and PbBr_2_ (1.8 M) in DMSO. The perovskite precursor solution was stirred at 75 °C overnight.

### 2.3. Device Fabrication

Planar inorganic perovskite solar cells were fabricated with a device structure of SLG/ITO/In_2_O_3_/CsPbI_2_Br/TFB/P3HT/Au. First 400 nm-thick ITO was deposited with a resistance of 5 Ω/sq by RF-sputtering method onto clean SLG substrate. Next, the 30 nm thin ETL was deposited at various growth temperatures by ALD onto the ITO substrate. And then, the CsPbI_2_Br perovskite precursor was spin-coated onto the substrates at 5000 rpm for 40 s. During spin-coating, hot air was blown to film using a hot air gun (BOSCH, GHG20–63) for 10 s (the hot air temperature was 300 °C). The perovskite films were annealed at 280 °C for 10 min on a hot plate. The substrates were transferred to a nitrogen-filled glove box for the subsequent deposition. TFB (0.5 mg mL^−1^ in CBZ) was spin-coated onto a perovskite film at 5000 rpm for 30 s and annealed at 100 °C for 5 min. P3HT (10 mg mL^−1^ in CBZ) as hole transport layer was spin-coated onto the perovskite light harvester at 3000 rpm for 30 s and then annealed at 100 °C for 5 min. Au electrodes were prepared by thermal evaporation by defining an effective cell area of 4 mm^2^ with a shadow mask.

### 2.4. Film Characterization

To analyze the property of In_2_O_3_ film, Si (for AFM, Ellipsometer, SEM, and XPS) and SLG (XRD, UV-visible, and Hall measurement) substrates were used. The thickness, refractive index, and optical bandgap of the ALD deposited In_2_O_3_ films were determined by an Ellipsometer (Model FS-1, Film sense, Lincoln, NE, USA, version: 1.69) with excellent thickness precision <0.001 nm. The measurements at 65° angle of incidence and the default refractive index were deduced at a single wavelength (λ) of 633 nm. The surface morphology was analyzed using scanning electron microscopy (SEM, JEOL, Tokyo, Japan, JSM-7610F) to accelerate voltage = 15 keV at room temperature with a secondary electron (SE) detector. The film’s topography and surface roughness were measured using an atomic force microscope (AFM, Pucotech, Seoul, Korea) under ambient conditions using noncontact mode. The crystallographic analysis of the ALD-In_2_O_3_ films was examined by X-ray diffraction (XRD, Rigaku diffractometer, Tokyo, Japan) using the 2θ method with CuKα radiation (λ = 1.5418 Å). XRD results were obtained from 2θ = 20–80° via step scan mode (step size of 0.02° and step time of 3 s). The chemical binding properties, film impurity, and O/In composition ratio were examined by X-ray photoelectron spectroscopy (PHI Quantera II, Seoul, Korea) after removing surface contaminants via ionized Ar sputtering and calibrated according to the C–C 1 s binding energy at 284.5 eV. The optical spectra of films were measured in a wavelength range from 300 to 1400 nm by UV-VIS spectrophotometer (Shimadzu, Kyoto, Japan, UV-2600). The electrical properties, like carrier concentration, mobility, and resistivity of the prepared films In_2_O_3_ on SLG, were measured by the Van der Pauw technique method by the Hall effect measurements system (Ecopia, Anyang, Korea, HMS-3000). First, on the SLG sample, indium probe was put in contact at all four edges and then loaded on a spring clipboard holder for use with the 0.51 Tesla magnet kit. The measurement was conducted at room temperature.

### 2.5. Device Characterization

The current-voltage (J-V) curves of the fabricated PSCs were measured using a solar simulator (PEC-L01, Peccell Technologies) under standard AM1.5 illumination (100 mW cm^−2^, 25 °C) in atmospheric air conditions.

## 3. Results

### 3.1. Film Growth Mechanism

In order to understand the reaction in the ALD chamber that occurs during the deposition process, first, we interpreted the possible reaction mechanism between indium precursor and oxidant. Typically, when Et_3_In and ozone molecules meet in the reaction chamber by sequential flow, they react to form the In_2_O_3_ layer. Four repeated steps form the ALD process, and the following growth can be divided into two reaction subcycles. Each exposure is accomplished during the ALD process by precursor adsorption self-limits of one monolayer of adsorbed precursor. In the first half-reaction, triethylindium molecules chemisorbed onto the substrate’s surface oxygen atoms; it was possibly split into free radicals of In* and organic species on the substrate surface (Equation (1)), which can be shown as below:(1)$$x{\mathrm{In}(C}_{2}H_{5}{)}_{3}\;\to \;x{\mathrm{In}}^{*}+3xC_{2}H_{5\;\left(g\right)}^{.}$$
* denote free radicals. The excess Et_3_In and gaseous byproducts should readily desorb from the surface and be washed off from the reaction chamber during Ar purge. Therein ozone decomposed to O_2_ and O*, from which it was acting as the oxidizer (Equation (2)).
(2)$$O_{3}\;\to \;O_{2\;}+O_{\;\left(g\right)}^{*}$$

The second half-reaction completed the process with the reacts of adsorbed species on the surface. Later Ar-purge evacuated the excess of all the byproducts. The overall ALD chemical reaction can be suggested as:(3)$$4\mathrm{In}{\left(C_{2}H_{5}\right)}_{3\;\left(g\right)}+3O_{3\;\left(g\right)}\;\to 2{\mathrm{In}}_{2}O_{3\;\left(S\right)}+xC_{n}H_{2n+2\;\left(g\right)}+yC_{n}H_{2n\;\left(g\right)}+H_{2}{\mathrm{CO}}_{3\;\left(g\right)}$$

The reaction during the production of In_2_O_3_ film and possible byproducts are mentioned in Equation (3). However, the reaction mechanism is not as simple as predict above. It could be a complex process due to steric hindrance caused by the bulky ligand molecules. Perhaps the ethyl groups cover reactive sites, and growth saturation is reached before every reactive site is occupied; meanwhile, the reaction kinetics directly depends on the provided activation energy.

### 3.2. Film Deposition: Thickness and Growth Rates

The self-limiting growth characteristics of Et_3_In and ozone surface reactions were determined by In_2_O_3_ film deposited at 150 °C and the pulse time of Et_3_In precursor was investigated. The growth per cycle (GPC) value was calculated by dividing the obtained film thickness by the number of cycles. Figure 1a shows a variation in the film GPC as a function of the Et_3_In precursor pulse time at the growth temperature of 150 °C. The ozone pulse time was fixed to 2 s, and the ALD cycles were 1000. A moderate variation in the film thickness might make possible due to substrates influence (error ± 2 nm). As shown here, the GPC increased with increasing Et_3_In dose time up to 0.3 s, and it was saturated with the GPC of 0.035 nm/cycle, and the additional precursor pulse time did not affect the film growth observed a relatively steady GPC. The saturation of the GPC confirmed that an ALD growth was achieved through self-limited adsorption of Et3In. We achieved the GPC with excellent reproducibility and uniformity of the In_2_O_3_ films for a timing sequence of 0.3:5:2:10 s at 150 °C. The ALD GPC was determined by the geometrical shape of the precursor molecules, the adsorbent’s density on the substrate surface and the saturation of adsorption overcome when several adsorbed atoms can be found by tightly packing the adsorbents on the surface sites [32]. Here, the deposition rate at 150 °C was much higher than the values obtained using the other indium precursors in previous studies: 0.027 nm/cycle at 200 °C [23], 0.029–0.033 nm/cycle at 150–300 °C [25], 0.010–0.016 nm/cycle at 100–250 °C [27]. It is noted that the optimized growth conditions for set-up the ALD growth directly depend on the appropriate design of the reactor, choice, and type of precursors, and also the type of oxidant; however, the precursor molecular size was generally larger than the oxidant molecules. Thus, if Et_3_In is used as a precursor for In_2_O_3_ growth, the oxidant’s nature may also influence the film growth rate. Furthermore, the obtained refractive index (Figure 1a) of the thin films was found to be ~1.92–1.98 at 632.8 nm measured by ellipsometry, which were acceptable valves at the same deposition condition.

In_2_O_3_ thin films were grown with different cycle numbers from 800 to 1300 cycles at 150 °C on Si substrates by above-optimized pulse condition. Figure 1b shows that very linear growth was observed, indicating self-limiting layer-by-layer growth behavior with validating GPC of 0.035 nm/cycle. To determine the effect of deposition temperature on the film growth, all In_2_O_3_ films were deposited for fixed 1000 ALD cycles with Et_3_In, and ozone pulse times were 0.3 and 2 s, at various growth temperatures from 115 to 250 °C, respectively. Figure 1c displays the growth rate of ALD-In_2_O_3_ thin films as a function of growth temperature using the optimized pulse conditions. Figure 1c shows that the growth rate was directly dependent on the deposition temperature, increasing with temperature with no separate temperature window within which a constant growth rate was observed, as reported earlier with other conditions [23,24]. Otherwise, our results are inconsistent with the previous reports; ALD-deposited In_2_O_3_ thin film with TMIn with ozone and TEIn with ozone was proposed in ALD process windows between 100 and 200 °C [29,31] and others [28]. Figure 1c shows that the growth rate of In_2_O_3_ increased drastically from 0.009 nm/cycle to 0.035 nm/cycle at a growth temperature of 115 to 150 °C. Abruptly, it rose 0.066 nm/cycle at 200 °C and 0.088 nm/cycle at 250 °C. The steady increase in GPC as the growth temperature increases can be explained by the thermal decomposition of Et_3_In [33]. The results indicated that In precursor does not thermally decompose between growth temperature ranges. It is noted that the thermolysis process enhanced at high temperatures to decompose radical species that rapid reactant adsorption took place and an increased film growth rate. Therefore, as temperature increases possibly enhanced the reactivity of surface-adsorbed species with ozone due to thermal activation of adsorption of Et_3_In [26]. Similar growth behavior results were observed in ALD-In_2_O_3_ thin films using the Me_3_In precursor with ozone reactant [29] at a growth temperature of 250 °C. Using these obtained GPC, In_2_O_3_ thin films were grown with different cycle numbers at 115, 200, and 250 °C temperatures on Si substrates. The In_2_O_3_ films thickness as a function of the number of cycles at various growth temperatures shown in Appendix A. A near-perfect linear fitting graph at all the growth temperatures, indicating precisely controlled layer-by-layer growth behavior exhibited during the ALD process, confirms the self-limiting process. The slope of the linear fitting procures a thickness of a film. As the growth temperature increased, the film thickness drastically enhanced in the following order: 115, 150, 200, and 250 °C. We could conclude that the increase in thickness is directly affected by the number of cycles applied at the appropriate temperature, confirming the ALD growth and GPC. 

### 3.3. Crystal Structure

The structure and crystallinity of ALD-In_2_O_3_ thin films grown at 115, 150, 200, and 250 °C were investigated via XRD, shown in Figure 2. For XRD analysis, ALD-In_2_O_3_ films were deposited in 1000 cycles with varied film thickness at 150–250 °C and 36 nm thin In_2_O_3_ film deposited at 115 °C, which has a similar thickness of 150 °C grown film. In_2_O_3_ films grown at 200 and 250 °C observed intense peaks at 2θ of 21.7°, 30.9°, 33.1°, 35.6°, 42.1°, 46.0°, 51.4°, and 61.0° assigned to the (211), (222), (123), (400), (332), (431), (440), and (622) planes, respectively. These were in good agreement with the standard JCPDS #44–1087 of the cubic phase of the In_2_O_3_ [29,34,35], and films showed the polycrystalline nature with a preferred orientation (222) plane. This kind of growth behavior has also been identified by previous ALD growth [25,26,27,31]. On the other hand, in In_2_O_3_ films deposited at a lower temperature, 115 °C, the crystalline structure of the thin film does not appear, indicating the film surface was entirely amorphous, since the peaks associated with the (222) plane were not observed at 115 °C, although (222) a plane peak appeared at 150 °C, revealing the amorphous with nanocrystalline nature of the film. Finally, the XRD result reveals that In_2_O_3_ films deposited at 115 °C were utterly amorphous, and the film grown at 150 °C had mixed features. 

The crystallite size (*D*) of the In_2_O_3_ thin films grown at 200 and 250 °C were calculated using the line broadening of the full width of half maximum (FWHM) of (222) prominent peak and the well-known Debye-Scherer formula (Equation (4)) [36].
(4)$$D=\frac{k\lambda }{\beta \cos\theta \;}\;$$
where *k* is a constant valve of 0.94, *λ* is the X-ray wavelength (0.154 nm), *β* is the FWHM of the preferred (222) peak, and θ is Bragg’s angle (in degrees). The grain size of In_2_O_3_ increased from 51.8 nm at 200 °C to 58.2 nm at 250 °C, showing a trend towards an upward direction to the growth temperature. Besides this, the deposition temperature increased from 200 °C to 250 °C, the relative intensity of the (222) diffraction peak increases rapidly, and the FWHM valve reduces to 0.018°. The grain size enhanced as the increased growth temperature has been attributed to the increase of nucleation density due to the high thermal effect and the thickness effect, owing to enhancing electron mobility. According to the XRD results, the film’s crystalline features are highly dependent on the growth temperature of the In_2_O_3_ films.

### 3.4. Surface Morphology

Scanning electron microscopy (SEM) was carried out to know the surface morphology of 30 nm-thick ALD-In_2_O_3_ films deposited at temperatures ranging from 115 to 250 °C. Here, in Figure 3, all prepared films were examined by SEM at high magnification (100,000). The SEM images (Figure 3a,b) show the films deposited at 115 and 150 °C had no visible structure or defects detected in the film. The film was smooth with congenital growth over the substrate’s entire surface, along with a few larger grain particles detected at 150 °C. However, high deposition temperatures of 200 °C and above showed an increase in grain size (40–45 nm) without visible pinholes, as displayed in Figure 3c,d. In addition, there were increased particle sizes, agglomeration, and microstructure due to the thermal energy effect. These findings were in agreement with the X-ray diffraction result. The In_2_O_3_ film grown at 200 and 250 °C deposition temperature showed polycrystalline features with plenty of larger scattered particles indicating a rough surface at the high growth temperature.

Figure 4a–d represents topography images of the 30 nm-thick ALD-In_2_O_3_ films grown at 115, 150, 200, and 250 °C, respectively, studied using atomic force microscopy (AFM). The root means square (RMS) roughness values of the films evaluated from the AFM images were 0.836, 1.664, 1.514, and 1.576 nm with the experimental errors ∆RMS = ±0.1 nm. Thus, the film was grown at 115 °C shows a significantly smooth surface and contained uniform grains with the lowest rough surface among all the deposited films, owing to predicting the amorphous phase. Although Figure 4b shows that the film has a coarse texture due to different sizes over the surface, as seen earlier in Figure 3b, which may be due to the phase transition from amorphous to polycrystalline starting at 150 °C, which might be a possible reason for the drastic increase in RMS roughness. Another possible reason could be that residual physisorbed species exists at the film surface. Eventually, as the growth temperature increases to 200 and 250 °C, these particles coalesced to form big grains, and the overall surface became uniform and smooth, which possesses a relatively slight decreased in roughness. Thus, the rough surface is ascribed to the influence of thermally decomposed surface species or improved crystallinity of the films, and the grain size becomes large, which is consistent with the SEM images.

### 3.5. Chemical Composition

The oxygen deficiency formed an n-type semiconductor oxide; hence, the oxygen vacancy concentrations in In_2_O_3_ play a significant role in the carrier concentrations of films and enhance the electrical property. The X-ray photoelectron spectroscopy (XPS) analysis was performed to investigate the chemical bonding properties, stoichiometric information, and the presence of impurities in the 30 nm-thick ALD-In_2_O_3_ films deposited on bare silicon substrates at 115, 150, 200, and 250 °C. Figure 5a presents the full survey spectrum, revealing In and O peaks that confirm indium oxide growth. Here, in our study, we use organometallic In precursor; therefore, carbon impurities might be present ingrown films. The C1*s* peak was examined in the carbon binding states; however, the C peak was not observed, revealing lesser impurities obtained in all prepared samples. For a detailed study, depth core level analysis of the deposited films was also examined.

Figure 5b–d shows the high-resolution XPS spectra of In3*d* (In3*d*_5/2_ and In3*d*_3/2_), O1*s*, and C1*s* core spectra, respectively, of ALD-In_2_O_3_ films using O_3_ plasma. Figure 5b shows the In3*d*_5/2_ peak observed at 444.3, 444.4, 444.1, and 444.1 eV at 115–250 °C, indicating In–O bond. The In3*d*_5/2_ binding energies valves are close to the reported In_2_O_3_ (444.2 eV) [23,37]. Compared to the binding energy of In_2_O_3_ film deposited at 150 °C, at high growth temperatures of 200 and 250 °C, the In3*d*_5/2_ core peak shifted slightly to the lower binding energy of 0.3 eV. In metal bonding, the binding energy peak appeared at 443.6 eV [37,38]; thus, the lower binding energy is attributed to the oxygen vacancy in the In_2_O_3_ films. The distance between the In3*d* (In3*d*_5/2_ and In3*d*_3/2_) spin-orbit peak to peak is 7.5 eV (taken from Figure 5b) confirms the materialization of In_2_O_3_ [37]. It is also shown that the indium possesses 3^+^ valency states in the In_2_O_3_ lattice structure in all the growth temperatures. Our findings are consistent with the previous reports for ALD-In_2_O_3_ using other precursors [24,25,29,30].

Figure 5c shows the high-resolution scan in the O1*s* binding energy states, and the peaks are deconvoluted into two subpeaks. The result is consistent with the In3*d* in which O–In (metal oxide binding energy 529–530 eV) was formed. A strong peak with a binding energy of 529.7 eV was observed at 115 and 150 °C, whereas, at high growth temperatures of 200 and 250 ℃, the O1*s* peak shifted towards slightly lower binding energy (0.1 eV), the same trend as In binding states observed. This high-intensity strong peak dovetails to the O bonds in the O–In lattice in the films [25,29,30,38]. Furthermore, this peak intensity increased along with growth temperature owing to the strong bonding and improved crystallinity of the grown films. Figure 5c shows that another oxygen peak of the XPS spectrum O1*s*, metal carbonates at 531.1–531.8 eV, was detected at all deposition temperatures. However, the peak intensity at lower growth temperatures was slightly higher compared to high deposition temperatures. This peak can be recognized as either surface or bulk hydroxyl–In species [24,29,38], perhaps originated from the incomplete formation of In_2_O_3_ lattice or the unreacted precursor molecules embedded into the film surface. The C1*s* core XPS spectra of the deposited films were examined and shown in Figure 5d. The C1*s* peaks were present in all the deposited temperatures, although it was slightly observed at the higher growth temperature present in all films and hard to glide by the ALD reaction process due to steric effect [23,28,29]. The reason could not be entirely thermal deposition of indium precursor. 

The atomic concentrations of In, O, and C in the ALD-In_2_O_3_ films at various deposition temperatures are displayed in Figure 5e. The film exhibited In (at.%) 37.74–39.28%, and O (at.%) 60.05–59.44% span in all grown temperatures. Our target is to develop device-quality impurity-free indium oxide films; henceforth, we ought to elaborate on the remaining surface residues in the films after the formation of In_2_O_3_. The possible impurity in the In_2_O_3_ films might be C core on the surface or embedded in the film due to the ethyl (-C_2_H_5_) molecule present in the indium precursor. The total carbon contamination is 2.2 at.% at 115 °C grown film, which was the highest impurity among all ALD deposited In_2_O_3_ films. The observed impurity can be ascribed to ethyl groups’ incomplete oxidation at relatively low deposition temperatures [23,24,25,29]. Regarding the carbon content in deposited films, C1*s* core persists 0.73, 0.64, 0.78, and 0.78 at.% as hydrocarbon (C–C and C–H; binding energy ~284.8 eV) film deposited at 115 to 250 °C, respectively. Another C peak (binding energy 289.5 eV), which can be assigned O–C=O (metal-carbonate), was also observed at 115 and 150 °C with 1.47 and 1.27 at.%, and at higher temperature (200 and 250 °C) carbonate impurity the figures decreased to 0.69 and 0.50 at.%, which also can be seen in Figure 5d as the peak is close to disappearing. Finally, almost negligible carbon was internalized 1.47 at.% in 200 °C and 1.28 at.% in 250 °C deposited films. The existing impurities might lead to a lower carrier concentration of the ALD-In_2_O_3_ thin films at low temperatures, and the carrier concentration drastically increased after removing the impurities. The noticeable point is that interpretation of the O/In ratio using XPS quantitative analysis is not highly precise despite considering the atomic sensitivity factors of all elements. The O/In ratio in ALD-grown films is presented in Figure 5f. The O/In ratio of 1.59 is at a low temperature, 115 °C, and 1.55 is at a higher temperature, 250 °C, determined from the compositional depth profile (Figure 5e). Generally, based on the core peaks obtained from In3*d* and O1*s*, the atomic ratio of indium oxide was nearly 1.5; thus, it can be concluded that the amounts of oxygen vacancy are more than that of crystal lattice O atoms. Each oxygen vacancy offers two electrons and an active site for the adsorption of oxygen species. Nevertheless, the differences between the O/In ratios indicate the remarkable phase controllability of the InO_x_ ALD processes. 

XPS analysis indicates that all films are In-enriched and have a higher density of oxygen vacancies, anticipating the n-type conductivity [17], which we analyzed via the Hall-effect and present in the next section. Our XPS results suggested that relatively pure In_2_O_3_ film was deposited at a higher temperature, ≥200 °C since the impurity was close to negligible (>1.4%), and the O/In ratios and the impurity level of the In_2_O_3_ films were not strongly affected by the growth temperature. The analysis clarifies by employing more purge during the ALD process, the detected impurities washed away from the reaction chamber and procured the high-quality film from 150 to 250 °C growth temperature, as below 150 °C, the deposited films may not be reliable for the device fabrication.

### 3.6. Optical Properties

The optical properties of the 30 nm-thick ALD-In_2_O_3_ grown at various temperatures are depicted in Figure 6a. All the fabricated films were highly transparent as 80–97% optical transmittance was in the visible region of 380–800 nm at all growth temperatures as mandatory for this TCO category material. Moreover, the transmittance of films at the wavelength of 550 nm was beyond 94%, suitable for a transparent electrode. The transmittance of films slightly decreased with the increased growth temperature. This lower transmittance at high deposition temperatures could be attributed to the film crystallization. However, 150 °C grown film showed a lower transmittance than 200 °C, ascribed to a slightly higher film roughness of 150 °C [24]. ALD-prepared 40 nm-thick In_2_O_3_ film using the Et_3_In precursor and O_3_ at the growth temperature range of 50–250 °C determined the optical transmission of over 80% in the wavelength range of 500–700 nm [31] and also our films’ transmittance values are similar with previous ALD deposited In_2_O_3_ [23,24,26,28,29,30]. The films’ optical transmittance is related to film crystallinity. Grain boundaries enlarged and the surface became slightly rough due to increased growth temperature.

The refractive index (R_F_) values of the prepared films deposited using a fixed number of 1000 ALD cycles at different growth temperatures (115–250 °C) were also extracted from the ellipsometry measurements at a wavelength of 632.8 nm. The refractive index increased from 1.79 to 2.03, increasing growth temperature from 115 to 250 °C (Figure 6 (inset)). Pure In_2_O_3_ has a refractive index (n) around 2.5 [39], and it corresponds to the density of the films. As the refractive index gives a crude insight into the films’ density, the low refractive index of films deposited at lower temperature suggests the films were less dense, which is obvious considering the film’s porous morphology discussed in the next paragraph. At the temperature of 115 °C, the refractive index value of 1.79 could be explained by incorporating impurities in the films or unreacted ligands or physisorbed species on the film surface. The chemical reactivity can enhance the formation of a dense, pinhole-free film because of higher thermal energy. Hence, at higher temperatures over 150 °C, the densification of films increased the refractive index.

The optical bandgap energy was extracted using the Tauc plot [40]. According to Equation (5), the bandgap has a direct relation with absorption coefficient and photon energy:(5)$$\alpha hv=A{\left(hv-E_{g}\right)}^{n}$$
where h is Planck’s constant, *ν* is photo frequency, *A* is a constant, *E_g_* is the optical bandgap, and *n* is equal to ½ and 2 for direct transitions and indirect transitions. For this, the fundamental absorption coefficient (*α*) of the In_2_O_3_ film was evaluated from Lambert’s law (Equation (6)) [41]
(6)$$\alpha =\frac{-\mathrm{Ln}(T/\left(1-R{)}^{2}\right)}{d}$$
where *d*, *T,* and *R* are the film thickness, transmittance, and reflection of the particular film, respectively. Figure 6b shows the variation in the ($\alpha hv{)}^{2}$ values of the In_2_O_3_ films as a function of photon energy. The nature of the plots indicates the existence of indirect optical transitions. The bandgap of the films was obtained by using the extrapolation of the linear portion of ($\alpha hv{)}^{2}$ to an intersection with the *x*-axis. Figure 6b shows that the In_2_O_3_ thin films exhibited bandgaps ranging from 3.82–3.75 eV. These values are consistent with reported values of In_2_O_3_ films deposited by ALD and other deposition methods [24,31]. On the other hand, In_2_O_3_ film deposited at 150 °C exhibits an optical bandgap at ~3.56 eV, similar to the bulk single-crystalline cubic In_2_O_3_ (Eg = 3.55 eV) [35]. The result, lower optical bandgap at 150 °C, contradicts the previous ALD-In_2_O_3_ report with the same indium precursor (triethylindium) and ozone as the oxidant [31], although it has a consistent ALD-In_2_O_3_ result with the trimethylindium and ozone precursor thin film deposited at 150 °C (Eg = 3.4 eV) [29]. The bandgap was narrowed due to electron-electron scattering and electron-impurity shielding effect in In_2_O_3_ films [35]; hence, the degeneracy in the carrier density in high oxide films effectively shifts the absorption edge towards the lower energy.

### 3.7. Electrical Properties

The 30 nm-thick In_2_O_3_ film was tested at room temperature using Hall Effect measurement to evaluate the electrical properties. Figure 7 shows the deposition temperature dependency on carrier concentration, resistivity, and mobility of In_2_O_3_ films fabricated at various temperatures. The ALD-In_2_O_3_ films exhibited a negative Hall coefficient indicating that all the In_2_O_3_ films show n-type characteristics because of intrinsic defects, such as O deficiencies and interstitial In defects [42]. Noticeably, the film’s electrical properties were observed towards the film deposition temperatures with controlled film thickness. The carrier concentration of ALD grown films increased gradually (3.9 × 10^15^–1.5 × 10^20^ cm^−3^) with growth temperature 115 to 200 °C and expectedly dropped to 2.6 × 10^19^ cm^−3^ at 250 °C. Nonetheless, the carrier concentration of the In_2_O_3_ film was more than for In_2_O_3_ bulk single crystals [6,43]. There is a similar trend observed in the film resistivity. Initially, at 115 °C, the film showed high electrical resistivity, 3.5 × 10^1^ Ω.cm, which rapidly dropped to 6.1 × 10^−3^ Ω.cm as the growth temperature increased to 150 °C. After that there was a slight reduction of 4.6 × 10^−3^ Ω.cm at 200 °C, although it showed 1.4 × 10^−2^ Ω.cm at the growth temperature of 250 °C. Thus, the film resistivity was more than the bulk single-crystal valve [43]. This shows that the growth temperature significantly affects the physical and chemical properties of the In_2_O_3_ films. The result reveals that the carrier concentration directly depends on the film stoichiometry composition and film purities, such as after removal of the impurities, the carrier concentration drastically increases, as confirmed from XPS examination (115–150 °C). The high carrier concentration and low resistivity could be primarily accounted for by scattering the electrons from ionized and neutral impurities and film crystallinity improvement to reduce the scattering originating from grain boundaries [44]. While the deposition temperature increases, thermal energy excited the free electrons from the valence band into the conduction band, and as a result, the forbidden energy gap expanded, which determined an increase in carrier concentration and conductivity as the substantial impurity prohibited the movements of the electrons, which had decreased the carrier concentration and conductivity. The high carrier concentrations probably enhanced charge transfer.

Figure 7 also shows a significant improvement in Hall mobility: 4.5, 9.9, 11.3 and 19.5 cm^2^ V^−1^ s^−1^ in the deposition temperatures of 115 to 250 °C, attributed to highly crystalline features from the amorphous ALD In_2_O_3_ films. The temperature dependence of mobility is related to the scattering mechanisms of free electrons and acoustical phonon scattering, a dominant mechanism in reactively sputtered polycrystalline In_2_O_3_ films, strongly degenerating [45]. The significant grain boundary scattering could ascribe the low Hall mobility at low temperature due to small grain structure and high grain boundary density, evidenced by the SEM examination, and the ionized impurity scattering is also the dominant mechanism. It also increased as the temperature rose because more extensive and denser grains were obtained at high deposition temperatures. This result can conclude that carrier concentration and Hall mobility are no longer independent of each other. Nevertheless, the high electron mobility of In_2_O_3_ film was a benefit to enhance the charge transfer and collection at the ETL/perovskite interface. Finally, compact, uniform, pinhole-free In_2_O_3_ film with remarkable optoelectronic properties can effectively transfer charge carriers from the perovskite and suppress charge recombination at the interface.

### 3.8. Photovoltaic Characteristics

Figure 8a illustrates a device schematic of the planar n-i-p configuration of ITO/ALD-In_2_O_3_/CsPbI_2_Br/TFB/P3HT/Au to examine the impact ALD deposited In_2_O_3_ as ETL resulting photovoltaic device performance. The solar cell device was fabricated on an ITO/glass substrate. The 30 nm-thick In_2_O_3_ films were deposited on the ITO substrates via the ALD method. The CsPbI_2_Br inorganic perovskite photoactive layer deposited on ETL followed by organic hole transport layer (HTL) was coated on the top of the perovskite, come after the top Au electrode deposition. Figure 8b shows the corresponding energy levels of each component of the PSC device. The valence band and conduction band value of In_2_O_3_ shown in Figure 8b is taken from the earlier report [10]. The energy band structure is expected to facilitate charge collection, guiding electrons and holes to move in opposite directions across the entire perovskite layer to the In_2_O_3_ electron transporting layer and TFB/P3HT hole transporting layer.

Most importantly, ALD grown In_2_O_3_ has never been used as ETL for PSCs. Therefore, to compare In_2_O_3_ as ETL, we used another already establishing ETL layer, sol-gel SnO_2_ [45], and fabricated the device with replaced the ALD-In_2_O_3_ instead of SnO_2_ based PSCs. Furthermore, to influence the effect of ALD deposition temperature on the device performance, we used In_2_O_3_ deposited at various growth temperatures (150 °C, 200 °C, and 250 °C). The J-V characteristics of the devices based on growth temperature-dependent In_2_O_3_ ETLs were measured under AM 1.5 illumination in atmospheric air conditions, and the resulting device parameters such as short-circuit current density (J_SC_), open-circuit voltage (V_OC_), fill factor (FF), and power conversion efficiency (η) are summarized in Table 1. The corresponding current density-voltage (J-V) curves are shown in Figure 8c.

The without In_2_O_3_ ETL, SnO_2_-controlled device exhibited a PCE of only 9.52%, with open-circuit voltage (V_OC_) of 1.12, short circuit current density (J_SC_) of 12.54 mAcm^−2^, and fill factor (FF) of 67.89% under the same condition. Contrarily, the modified ETL to ALD-In_2_O_3_ can effectively improve the device’s performance. Figure 8c clearly shows that the PCE was enhanced, and the best performance with ALD-In_2_O_3_ was 250 °C grown film, exhibited JSC of 13.64 mA/cm^2^ FF of 70.99%, and V_OC_ of 1.13 V, which yielded remarkable PCE of 10.97%. In contrast, the PSC fabricated with 200 °C grown ALD-In_2_O_3_ film has a low J_SC_ of 12.13 mA/cm^2^, FF of 68.54%, and a slight high V_OC_ of 1.15 V, and the lowest PCE of 9.53%. Although the In_2_O_3_ film deposited at 150 °C displays the improved J_SC_ of 13.18 mA/cm^2^, dropped FF of 67.37%, and V_OC_ of 1.10 V, resulting in the PCE of 9.77%. The reduced V_OC_ and FF observed in PSC fabricated with 150 °C ALD-In_2_O_3_ indicated the recombination is increased at the ETL/Perovskite interface.

Compared to the 250 °C In_2_O_3_ ETL device, PSCs based on 150 °C and 200 °C ALD- In_2_O_3_ exhibited low J_SC_. Usually, the high conductivity of ETL may cause current leakage, and may this could be a possible reason for the drop of J_SC_ at ≤200 °C. Whereas, at 250 °C, increased J_SC_ is mainly attributed to an enhanced electron extraction at the ETL/perovskite interface, including reduced charge recombination contributing to remarkably enhanced FF [46].

The film quality of In_2_O_3_ is critical for charge transfer and collection, and the result suggested that the ALD-In_2_O_3_ film is much more efficient than conventional solution-processed SnO_2_ nanoparticles ETL. The device result indicated that the solar cell performance directly depends on the film morphology, crystallinity, high mobility, and wide bandgap and electrical properties of In_2_O_3_ films that contribute to the performance improvement of PSCs. High-quality In_2_O_3_ ETLs efficiently transferred charge carriers and suppressed change recombination, improving the J_SC_, FF, and PCE, unless V_OC_ is higher in 200 °C deposited In_2_O_3_ based devices. The low V_OC_ of the device (250 °C ALD-In_2_O_3_) was mainly attributed to the charge recombination, which led to the V_OC_ loss [13]. The 120 nm-thick and 40 nm-thick In_2_O_3_ films made by sol-gel method followed by annealing at 200 °C with fabricated organic-inorganic hybrid-type PSCs achieved a PCE of 13.01% [10] and 15.3% [12]. In addition, we estimated the hysteresis index (HI) values as indicated in Appendix A. Only the In_2_O_3_ sample made at 200 °C exhibited the HI value similar to the control device using solution-processed SnO_2_ ETL. On the other hand, other In_2_O_3_ samples made at 150 and 250 °C revealed a considerably decreased HI, ascribed to the relatively compact structure of ALD In_2_O_3_ thin film. Our best condition of In_2_O_3_ made at 250 °C meets the highest PCE and lowest HI value, surpassing conventional solution-based SnO_2_ ETL.

Our PSCs device result is comparable to previous reports as here we used inorganic perovskite materials, which have more stability. Overall, the present work provided new insight into ALD-In_2_O_3_ can be a promising ETL material for advancing planar PSCs’ performance.

## 4. Conclusions

Low-temperature-processed, device-quality In_2_O_3_ thin films were successfully deposited by the ALD method. These have been fully characterized and mobilized as ETL for planar PSCs. The self-limiting phenomena through the nucleation and film properties were significantly dependent on the deposition temperature. It is worth observing that the film GPC value steeply increased from 0.035 nm/cycle to 0.088 nm/cycle between the growth temperatures of 150 and 250 °C due to the rapid dissociation of Et_3_In precursor. An amorphous phase turned to the polycrystalline cubic at ≥200 °C, including a high transmittance of >81–97% in the visible region obtained at all the growth temperatures. At 200 °C and above, the film obtained high conductivity with reduced resistivity (4.6 × 10^−3^ and 1.4 × 10^−2^ Ω·cm) and high carrier concentration (1.5 × 10^20^ and 2.6 × 10^19^ cm^−3^). The compact, pinhole-free In_2_O_3_ film with remarkable optoelectronic properties could effectively transfer charge carrier from the perovskite and suppressed charge recombination at the interface. The ALD In_2_O_3_ layer was deposited at 250 °C while the fabricated PSCs exhibited the J_SC_, FF, and PCE’s best device performance as 13.64 mA/cm^2^, 70.99%, and 10.97%, respectively. The above findings corroborate that In_2_O_3_ can be a possible alternative material for basic ETL to achieve efficient perovskite solar cells.

## Figures and Tables

**Figure 1 nanomaterials-11-02047-f001:**
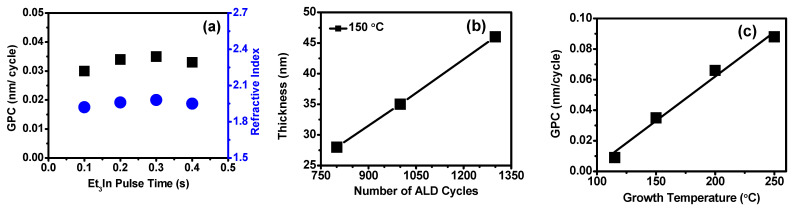
(**a**) The variation in the growth rate (■) and refractive index (●) of In_2_O_3_ films as a function of the Et_3_In precursor pulse time at 150 °C (**b**) In_2_O_3_ film thickness as a function of the number of ALD cycles for the film deposited at 150 °C on the Si substrate (**c**) Variation in In_2_O_3_ film GPC at various growth temperatures.

**Figure 2 nanomaterials-11-02047-f002:**
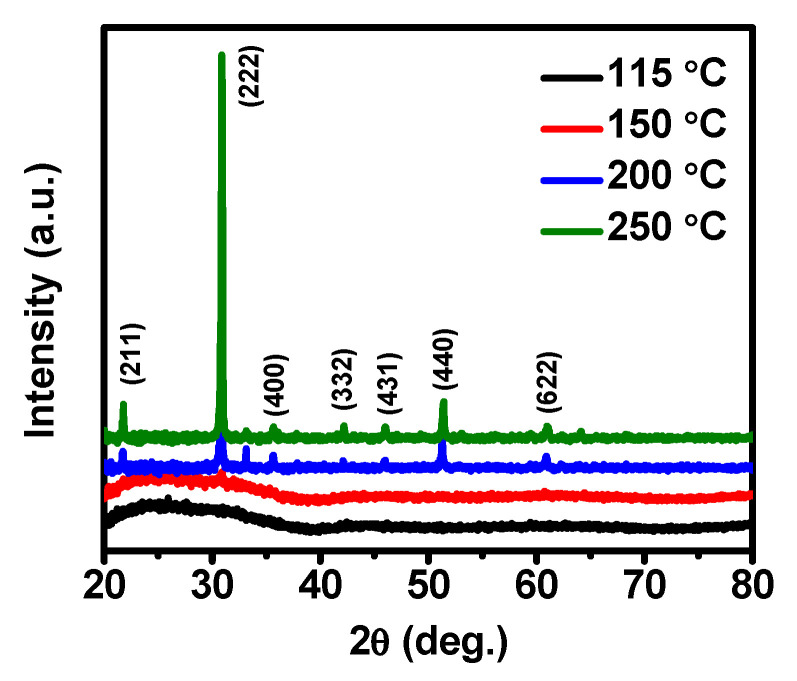
XRD patterns of ALD-In_2_O_3_ films deposited on various growth temperatures.

**Figure 3 nanomaterials-11-02047-f003:**
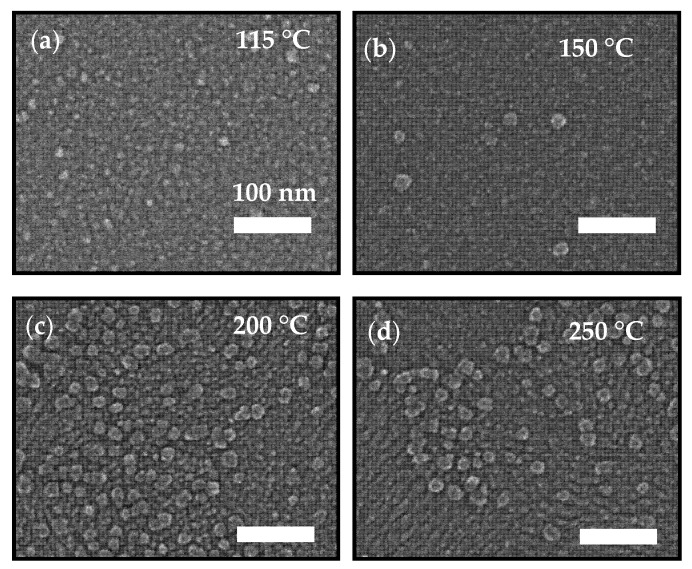
Plain-view SEM images of In_2_O_3_ films deposited on the Si substrate at (**a**) 115 °C (**b**) 150 °C, (**c**) 200 °C, and (**d**) 250 °C. All scale bars are 100 nm.

**Figure 4 nanomaterials-11-02047-f004:**
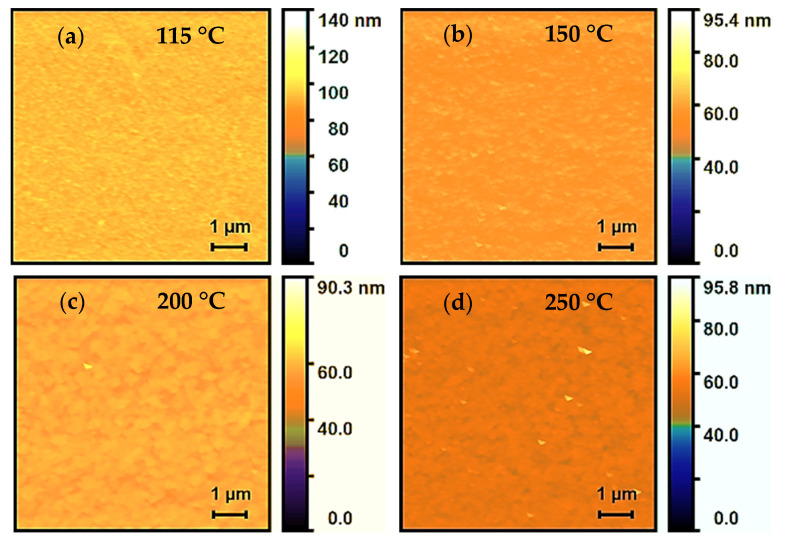
AFM images of In_2_O_3_ films deposited at (**a**) 115 °C, (**b**) 150 °C, (**c**) 200 °C, and (**d**) 250 °C.

**Figure 5 nanomaterials-11-02047-f005:**
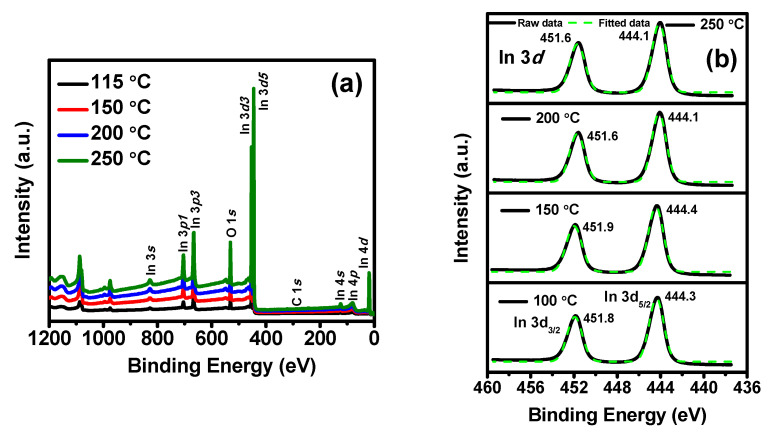
(**a**) Full survey XPS spectra of ALD prepared In_2_O_3_ films grown at 115 °C, 150 °C, 200 °C, and 250 °C; high-resolution XPS spectra of (**b**) In 3*d* core level, (**c**) O 1*s* core level, (**d**) C 1*s* core level, and (**e**) XPS elemental depth profiles (at.%). (**f**) [O/In] ratio of In_2_O_3_ films grown at 115–250 °C.

**Figure 6 nanomaterials-11-02047-f006:**
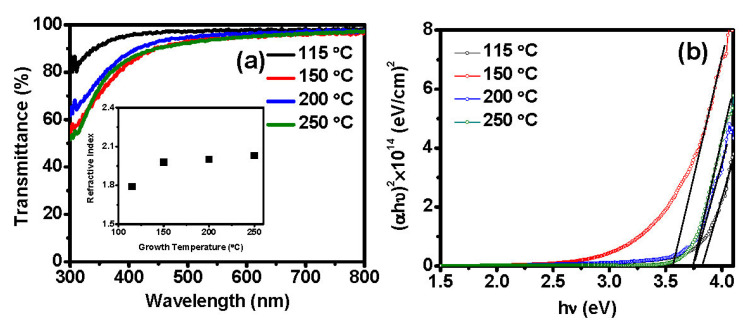
(**a**) The transmittance spectra with (inset) the variation of refractive index for the films deposited at 115 °C, 150 °C, 200 °C, and 250 °C with 1000 ALD cycles, (**b**) Tauc plots of (αhν)^2^ vs. hν for extracting the optical bandgap energy and (inset) the extracted bandgap energy of In_2_O_3_ films grown at 115 °C, 150 °C, 200 °C, and 250 °C.

**Figure 7 nanomaterials-11-02047-f007:**
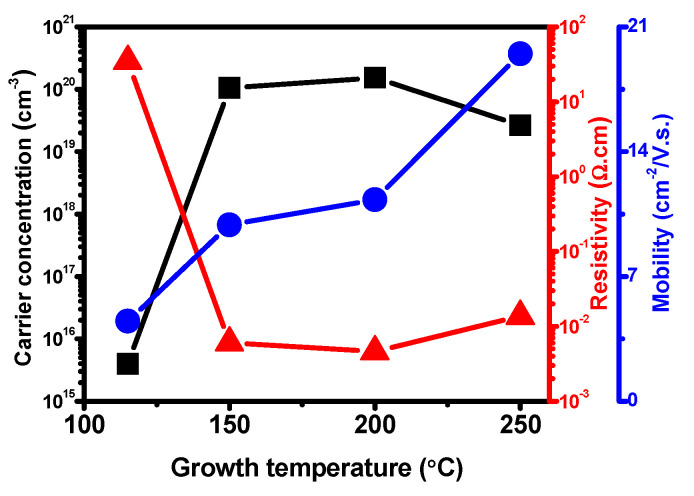
The variations in carrier concentration (■), resistivity (▲), and Hall mobility (●) of the In_2_O_3_ films deposited on a glass substrate as a function of the growth temperatures (115 °C, 150 °C, 200 °C, and 250 °C).

**Figure 8 nanomaterials-11-02047-f008:**
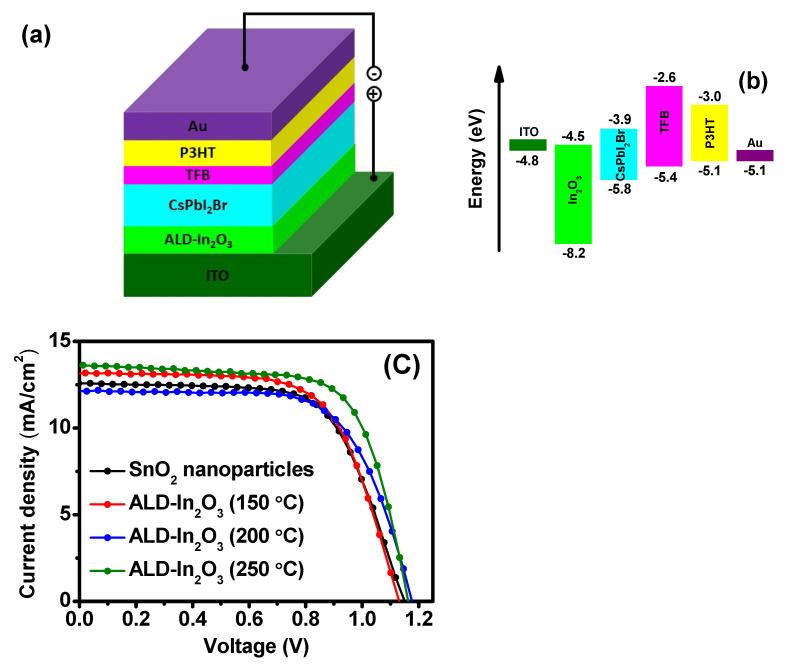
(**a**) Schematic of the device structure of the inorganic perovskite solar cell and (**b**) energy level diagram (**c**) J-V characteristics of CsPbI_2_Br PSCs with SnO_2_ nanoparticles, ALD-In_2_O_3_ (deposited at 150 °C, 200 °C, and 250 °C).

**Table 1 nanomaterials-11-02047-t001:** Photovoltaic parameters of CsPbI_2_Br PSCs with SnO_2_ nanoparticles ETL and ALD-In_2_O_3_ ETLs deposited at 150 °C, 200 °C, and 250 °C under AM1.5 illumination.

ETL	V_OC_ [V]	J_SC_ [mA cm^−2^]	FF [%]	PCE [%]
SnO_2_	1.12	12.54	67.89	9.52
In_2_O_3_ 150 °C	1.10	13.18	67.37	9.77
In_2_O_3_ 200 °C	1.15	12.13	68.54	9.53
In_2_O_3_ 250 °C	1.13	13.64	70.99	10.97

## Data Availability

Not applicable.

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
