# Peer review of "Growth Temperature Influence on Atomic-Layer-Deposited In2O3 Thin Films and Their Application in Inorganic Perovskite Solar Cells"

_nanomaterials, 2021, doi:10.3390/nano11082047_

Round 1
Reviewer 1 Report
In this paper, the authors report on the preparation and characterization of solution-processed In2O3 layers for perovskite solar cells. In2O3 or other metal oxide layers are very important in perovskite solar cells as electron transport layers (ETL), and low temperature process fabrication of ETL layers is one of the current extensive research in perovskite solar cells. In this paper, the authors demonstrate the atomic layer deposition (ALD) of In2O3 manipulation in different temperature ranges. They show that the morphology of In2O3 is affected by the process temperature and that this morphology is related to the photovoltaic performance of the device.
While I believe this paper will attract a wide audience and basically meets the requirements for publication, I would like to comment on the details of the experimental results.
- Why was the CsPbI2Br perovskite chosen for this experiment? Was it because the band matching between CsPbI2Br and In2O3 was appropriate?
- For the J-V curve, what is the hysteresis index between the forward and reverse scans of the J-V curve? Was the hysteresis index unaffected by the preparation conditions of In2O3?
Author Response
Response to Reviewer 1 Comments
In this paper, the authors report on the preparation and characterization of solution-processed In2O3 layers for perovskite solar cells. In2O3 or other metal oxide layers are very important in perovskite solar cells as electron transport layers (ETL), and low temperature process fabrication of ETL layers is one of the current extensive research in perovskite solar cells. In this paper, the authors demonstrate the atomic layer deposition (ALD) of In2O3 manipulation in different temperature ranges. They show that the morphology of In2O3 is affected by the process temperature and that this morphology is related to the photovoltaic performance of the device.
While I believe this paper will attract a wide audience and basically meets the requirements for publication, I would like to comment on the details of the experimental results.
Thanks so much for reviewing our research paper, titled "Growth temperature influence on atomic layer deposited In2O3 thin films and its application for inorganic perovskite solar cells". We have carefully looked into the reviewer's excellent suggestions to improve our manuscript and made corrections accordingly. The followings are the reply to the reviewer's comments:
Point 1: Why was the CsPbI2Br perovskite chosen for this experiment? Was it because the band matching between CsPbI2Br and In2O3 was appropriate?
Response 1: Thank you for the reviewer's thoughtful comment.
As the reviewer pointed out, we have chosen the CsPbI2Br perovskite for light-harvesting material because it provides appropriate energy band matching with the In2O3 electron transport layer.
Point 2: For the J-V curve, what is the hysteresis index between the forward and reverse scans of the J-V curve? Was the hysteresis index unaffected by the preparation conditions of In2O3?
Response 1: Thank you for the reviewer's important comment.
As the reviewer suggested, we have estimated the HI (hysteresis index) values as indicated in Figure S2 and Table S1. In addition, on page 15, we have added the hysteresis index and modified the discussion paragraph in the Photovoltaic Characteristics section.
Only In2O3 sample made at 200 oC exhibited HI value similar to control device using solution-processed SnO2 ETL. On the other hand, other In2O3 samples made at 150 and 250 oC revealed a much improved HI value of 0.85, which can be ascribed to the relatively compact structure of ALD In2O3 thin film. Our best condition of In2O3 made at 250 oC meets the highest PCE and lowest HI value, which surpasses conventional solution-based SnO2 ETL.
Figure S2. Hysteresis observation for developed CsPbI2Br perovskite solar cells with different ETLs.
Table S1. Photovoltaic parameters and hysteresis index of fabricated CsPbI2Br perovskite solar cells with different ETLs.
|
Sample |
JSC (mA/cm2) |
VOC (V) |
FF (%) |
PCE (%) |
Hysteresis Index (HI) |
|
SnO2 reverse scan |
12.54 |
1.12 |
67.89 |
9.52 |
0.75 |
|
SnO2 forward scan |
12.62 |
1.11 |
50.76 |
7.11 |
|
|
In2O3 150℃ reverse scan |
13.18 |
1.10 |
67.37 |
9.77 |
0.85 |
|
In2O3 150℃ forward scan |
13.41 |
1.07 |
58.06 |
8.33 |
|
|
In2O3 200℃ reverse scan |
12.13 |
1.15 |
68.54 |
9.53 |
0.74 |
|
In2O3 200℃ forward scan |
12.92 |
1.07 |
50.71 |
7.01 |
|
|
In2O3 250℃ reverse scan |
13.64 |
1.13 |
70.99 |
10.97 |
0.85 |
|
In2O3 250℃ forward scan |
13.75 |
1.07 |
62.99 |
9.27 |

Reviewer 2 Report
This manuscript explores the growth temperature-dependent properties of In2O3 thin film fabricated by ALD using the triethyl indium (Et3In) precursor, and the corresponding results confirm that the structural, optical and electrical properties of the films directly depend on film growth temperature. Furthermore, the CsPbI2Br PSCs employing ALD-In2O3 films deposited at 250 °C as ETLs exhibit a champion PCE of 10.97%. However, this manuscript is not well organized, and there are several issues that need to be addressed. Thus, I would recommend major revision of this current manuscript.
- The manuscript is lack of novelty. In the previous report (Journal of Alloys and Compounds 649 (2015) 216-221), Maeng et al. successfully prepared indium oxide films by ALD using three different liquid precursors and ozone as the reactant, and the electrical, structural and optical properties of films were systematically investigated as functions of the deposition temperature and precursors. Undoubtedly, the related report in 2015 remarkably reduces the novelty of this work.
- The authors declare that "The temperature effect caused the increase of RMS roughness and because of that, the In2O3 films crystalline grains enlarged". There is no direct relationship between RMS roughness and grain size. Please give a reasonable and more specific explanation for this conclusion.
- The authors point out that "The transmittance of films slightly decreased with the increased growth temperature, except that 150 °C grown film showed the lowest transmittance". Please give a reasonable explanation and analysis for the lowest transmittance of 150 °C tailored sample.
- How did the authors obtain the carrier concentration, mobility and resistivity of the In2O3 films? Please provide more characterization details.
- For Figure 8b, how did the authors get the valence band and conduction band of the ALD-In2O3 film? UPS? Please provide more evidence. In addition, what kind of ALD-In2O3 film exhibits the band gap of 3.7 eV?It is difficult to find the relevant information in the manuscript.
- Please unify the writing format of Fig or Figure in the manuscript.
Author Response
Response to Reviewer 2 Comments
This manuscript explores the growth temperature-dependent properties of In2O3 thin film fabricated by ALD using the triethyl indium (Et3In) precursor, and the corresponding results confirm that the structural, optical and electrical properties of the films directly depend on film growth temperature. Furthermore, the CsPbI2Br PSCs employing ALD-In2O3 films deposited at 250 °C as ETLs exhibit a champion PCE of 10.97%. However, this manuscript is not well organized, and there are several issues that need to be addressed. Thus, I would recommend major revision of this current manuscript.
Thanks so much for reviewing our research paper, titled "Growth temperature influence on atomic layer deposited In2O3 thin films and its application for inorganic perovskite solar cells". We have carefully looked into the reviewer's excellent suggestions to improve our manuscript and made corrections accordingly. The followings are the reply to the reviewer's comments:
Point 1: The manuscript is lack of novelty. In the previous report (Journal of Alloys and Compounds 649 (2015) 216-221), Maeng et al. successfully prepared indium oxide films by ALD using three different liquid precursors and ozone as the reactant, and the electrical, structural and optical properties of films were systematically investigated as functions of the deposition temperature and precursors. Undoubtedly, the related report in 2015 remarkably reduces the novelty of this work.
Response 1: It is the fact that first-time Maeng et al. already reported in Journal of Alloys and Compounds 649 (2015) 216-221, about triethylindium precursor and ozone as the reactant along with two others In precursor, and the electrical, structural, and optical properties of films were systematically investigated as functions of the deposition temperature and precursors. This work we already mentioned in our manuscript as reference [31]. The report studied 40 nm thick In2O3 film at the growth temperature range of 50 to 250 °C. Here we study 30 nm thick In2O3 thin film at the deposition temperature of 115 to 250 °C. Moreover, some of our growth findings are different from previous work. For example, we found that as growth temperature increased, the film growth rate increased accordingly, while earlier report mentioned an ALD process window between 100 and 200 °C, where the GPCs remain constant. Further, the first time we employed the ALD grown In2O3 as an electron transport layer for inorganic planar perovskite solar cells.
Point 2: The authors declare that "The temperature effect caused the increase of RMS roughness and because of that, the In2O3 films crystalline grains enlarged". There is no direct relationship between RMS roughness and grain size. Please give a reasonable and more specific explanation for this conclusion.
Response 2: Thanks for the comment. On page 8, we have revised the conclusion with a better specific explanation on the surface morphology section.
Point 3: The authors point out that "The transmittance of films slightly decreased with the increased growth temperature, except that 150 °C grown film showed the lowest transmittance". Please give a reasonable explanation and analysis for the lowest transmittance of 150 °C tailored sample.
Response 3: We have revised and modified on page 11 in the Optical properties section.
Point 4: How did the authors obtain the carrier concentration, mobility and resistivity of the In2O3 films? Please provide more characterization details.
Response 4: On page 4, in the Film Characterization section, we have added the details of the electrical properties characterization.
Point 5: For Figure 8b, how did the authors get the valence band and conduction band of the ALD-In2O3 film? UPS? Please provide more evidence. In addition, what kind of ALD-In2O3 film exhibits the band gap of 3.7 eV?It is difficult to find the relevant information in the manuscript.
Response 5: Figure 8(b) shows the proposed energy band diagram of the device. The mentioned valence and conduction band value is taken from reference [10], and it is just a reference valve. We did not do a UPS study. On page 14, in the Photovoltaic Characteristics section, we have revised and added a reference to clarify.
Point 6: Please unify the writing format of Fig or Figure in the manuscript.
Response 6: We have revised the writing format with "Figure" in the manuscript. Thanks for your keen observation.

Reviewer 3 Report
The manuscript reported “Growth Temperature Influence on Atomic Layer Deposited In2O3 thin films and its Application for Inorganic Perovskite Solar Cells”. The detail of the ALD process at 115 – 250 °C has been studied through the film growth rate, crystal structure, morphology, composition, optical and electrical properties.. The method of material synthesis and characterization described in the manuscript are technically correct, and the data analysis are convincing. Nevertheless, the following comments/concerns should be addressed:
1.English needs to be polished in the manuscript.
- Someimages are not distinct.
3..There are some important previous works of perovskite materiles (Pseudohalide anion engineering for highly efficient and stable perovskite solar cells,Matter, 2021, 4, 1755–1767; Self-assembled monolayers in perovskite solar cells, Journal of Semiconductors, 2021, 42(9): 090202.; Lead-free halide double perovskite materials: a new superstar towards green and stable optoelectronic applications, Nano-Micro Letters, 2019, 11,16) should be cited.
Author Response
Response to Reviewer 3 Comments
The manuscript reported "Growth Temperature Influence on Atomic Layer Deposited In2O3 thin films and its Application for Inorganic Perovskite Solar Cells". The detail of the ALD process at 115 – 250 °C has been studied through the film growth rate, crystal structure, morphology, composition, optical and electrical properties. The method of material synthesis and characterization described in the manuscript are technically correct, and the data analysis are convincing. Nevertheless, the following comments/concerns should be addressed:
Thanks so much for reviewing our research paper, titled "Growth temperature influence on atomic layer deposited In2O3 thin films and its application for inorganic perovskite solar cells". We appreciate for taking the time to read and to providing valuable feedback on our manuscript. We have carefully looked into the reviewer's excellent suggestions to improve our manuscript and made corrections accordingly. The followings are the reply to the reviewer's comments:
Point 1: English needs to be polished in the manuscript.
Response 1: Regarding English polish, the manuscript has been revised to correct the sentence clearance as well as grammar and expression errors.
Point 2: Some images are not distinct.
Response 2: Thanks very much for the suggestion. As there is not any specific suggestion, so we have revised Figure 4 for more recognizable images.
Point 3: There are some important previous works of perovskite materials (Pseudohalide anion engineering for highly efficient and stable perovskite solar cells,Matter, 2021, 4, 1755–1767; Self-assembled monolayers in perovskite solar cells, Journal of Semiconductors, 2021, 42(9): 090202.; Lead-free halide double perovskite materials: a new superstar towards green and stable optoelectronic applications, Nano-Micro Letters, 2019, 11,16) should be cited.
Response 3: Thanks much for the reviewer's suggestion for three previous works cited in our manuscript. The first research work we found on pages 1762 – 1764 is very recent (Pseudohalide anion engineering for highly efficient and stable perovskite solar cells, Matter, 2021). However, the second research work (Self-assembled monolayers in perovskite solar cells, Journal of Semiconductors, 2021, 42(9): 090202), did not find it correctly. The last recommended research work is a review paper on "Lead-free perovskite materials" (Lead-free halide double perovskite materials: a new superstar towards green and stable optoelectronic applications, Nano-Micro Letters, 2019, 11,16).
Although these suggested reports (1 and 3) are very informative, we respectfully disagree with cite suggested works. Our study focuses on the inorganic perovskite materials, and that we already cited suitable references. Therefore, we believe these reports are out of the scope of our manuscript. Furthermore, our focus is on the environmentally reliable perovskite energy harvesting device rather than the lead-free device. Thank you.
Round 2
Reviewer 2 Report
The authors have responded to all comments the reviewer raised. While, some issues in the current version of manuscript still need to be addressed. Therefore, I would recommend publication of this work in Nanomaterials after addressing the following issues.
- Figure 6(b) shows that the 510 In2O3 thin films exhibited bandgap ranging from 3.82-3.75 eV. However, the bandgap of In2O3 in energy level diagram of CsPbI2Br PSCs, as shown in Figure 6(b), is 3.7 eV. This is obviously unreasonable and seems contradictory. In addition, I do not agree with the authors on the explanation that the mentioned valence and conduction band value of In2O3 is taken from reference [10], because the emphasis of this manuscript is on the successful preparation of In2O3 film by ALD and its application in PSCs as the ETL. Thus, I think it is necessary to determine the valence band and conduction band of In2O3 fabricated in this work via UPS.
- How did the authors obtain the hysteresis index (HI) of various PSCs? To my knowledge, this seems to be incorrect. The authors should recalculate the HI of PSCs according to relevant references.
Author Response
Response to Reviewer 2 Comments
The authors have responded to all comments the reviewer raised. While, some issues in the current version of manuscript still need to be addressed. Therefore, I would recommend publication of this work in Nanomaterials after addressing the following issues.
Thanks so much for reviewing our research paper, titled "Growth temperature influence on atomic layer deposited In2O3 thin films and its application for inorganic perovskite solar cells". Further, we have carefully looked into the reviewer suggestions, and here is the answer to the reviewer's comments:
Point 1: Figure 6(b) shows that the 510 In2O3 thin films exhibited bandgap ranging from 3.82-3.75 eV. However, the bandgap of In2O3 in energy level diagram of CsPbI2Br PSCs, as shown in Figure 6(b), is 3.7 eV. This is obviously unreasonable and seems contradictory. In addition, I do not agree with the authors on the explanation that the mentioned valence and conduction band value of In2O3 is taken from reference [10], because the emphasis of this manuscript is on the successful preparation of In2O3 film by ALD and its application in PSCs as the ETL. Thus, I think it is necessary to determine the valence band and conduction band of In2O3 fabricated in this work via UPS.
Response 1: Thank you for your comment. We agree that Figure 6(b) shows the In2O3 thin films exhibited bandgap ranging from 3.82-3.75 eV. Furthermore, Figure 8(b) displays the bandgap of In2O3 in the energy level diagram of CsPbI2Br PSCs shown 3.7eV that we mentioned taken from reference [10].
A previous report mentioned that the In2O3 optical bandgap is 3.75 eV, and the energy level diagram shows 3.7 eV [R1]. As well as another recent report showed that the optical bandgap of In2O3 is 3.81 eV, and the energy level diagram shows 3.7 eV [R2].
We predict that after UPS measurement, the determined bandgap of In2O3 could be the same or might be ±0.1 eV. Therefore, the UPS analysis is a good suggestion, but it will not change the In2O3 valence and conduction band value. Thank you for the suggestion of the UPS measurement.
The emphasis of the manuscript is on the successful preparation of In2O3 film by ALD at various growth temperatures and different growth temperatures deposited ALD-In2O3 application in PSCs as the ETL. Thank you for your valuable time.
R1. Qin, M.; Ma, J.; Ke, W.; Qin, P.; Lei, H.; Tao, H.; Zheng, X.; Xiong, L.; Liu, Q.; Chen, Z.; Lu, J.; Yang, G.; Fang, G. Perovskite solar cells based on low-temperature processed indium oxide electron selective layers. ACS Appl. Mater. Interfaces 2016, 8, 8460 – 8466. https://doi.org/10.1021/acsami.5b12849 (reference 10 in manuscript).
R2. Yang, B.; Ma, R.; Wang, Z.; Ouyang, D.; Huang, Z.; Lu, J.; Duan, X.; Yue, L.; Ning, X.; Choy, W.C.H. Efficient gradient potential top electron transport structures achieved by combining an oxide family for inverted perovskite solar cells with high efficiency and stability. ACS Appl. Mater. Interfaces 2021, 13, 27179 – 27187.
https://doi.org/10.1021/acsami.1c05284
Point 2: How did the authors obtain the hysteresis index (HI) of various PSCs? To my knowledge, this seems to be incorrect. The authors should recalculate the HI of PSCs according to relevant references.
Response 2:
Thank you for the reviewer’s important comment.
We had obtained the HI based on the general formula of PCEFOR/PCEREV. However, the reviewer may mean another HI estimation like HI=(PCEREV-PCEFOR)/PCEREV [R3]. Therefore, according to the reviewer’s suggestion, we have again estimated the HI values as indicated in Table S1.
Our best condition of In2O3 made at 250 oC meets the highest PCE and low HI value, surpassing conventional solution-based SnO2 ETL.
Thank you so much for the kind comment.
R3. Zhu Ma, Weiya Zhou, Zheng Xiao, Hua Zhang, Zhuoyun Li, Jia Zhuang, Changtao Peng, Yuelong Huang, Negligible hysteresis planar perovskite solar cells using Ga-doped SnO2 nanocrystal as electron transport layers, Organic Electronics, 71, 98-105. https://doi.org/10.1016/j.orgel.2019.05.011
Table S1. Photovoltaic parameters and hysteresis index of fabricated CsPbI2Br perovskite solar cells with different ETLs.
|
Sample |
JSC (mA/cm2) |
VOC (V) |
FF (%) |
PCE (%) |
Hysteresis Index (%) |
|
SnO2 reverse scan |
12.54 |
1.12 |
67.89 |
9.52 |
25.3 |
|
SnO2 forward scan |
12.62 |
1.11 |
50.76 |
7.11 |
|
|
In2O3 150℃ reverse scan |
13.18 |
1.10 |
67.37 |
9.77 |
14.7 |
|
In2O3 150℃ forward scan |
13.41 |
1.07 |
58.06 |
8.33 |
|
|
In2O3 200℃ reverse scan |
12.13 |
1.15 |
68.54 |
9.53 |
26.4 |
|
In2O3 200℃ forward scan |
12.92 |
1.07 |
50.71 |
7.01 |
|
|
In2O3 250℃ reverse scan |
13.64 |
1.13 |
70.99 |
10.97 |
15.5 |
|
In2O3 250℃ forward scan |
13.75 |
1.07 |
62.99 |
9.27 |
Fig. S2. Hysteresis observation for developed CsPbI2Br perovskite solar cells with different ETLs.
